# DSMNN-Net: A Deep Siamese Morphological Neural Network Model for Burned Area Mapping Using Multispectral Sentinel-2 and Hyperspectral PRISMA Images

Seyd Teymoor Seydi [1], Mahdi Hasanlou [1,*] and Jocelyn Chanussot [2,3]

1    School of Surveying and Geospatial Engineering, College of Engineering, University of Tehran, Tehran 14174-66191, Iran; seydi.teymoor@ut.ac.ir
2    Aerospace Information Research Institute, Chinese Academy of Sciences, Beijing 100094, China; jocelyn.chanussot@grenoble-inp.fr
3    CNRS, Grenoble INP, GIPSA-Lab, Université Grenoble Alpes, 38000 Grenoble, France
*    Correspondence: hasanlou@ut.ac.ir; Tel.: +98-21-6111-4525

**Abstract:** Wildfires are one of the most destructive natural disasters that can affect our environment, with significant effects also on wildlife. Recently, climate change and human activities have resulted in higher frequencies of wildfires throughout the world. Timely and accurate detection of the burned areas can help to make decisions for their management. Remote sensing satellite imagery can have a key role in mapping burned areas due to its wide coverage, high-resolution data collection, and low capture times. However, although many studies have reported on burned area mapping based on remote sensing imagery in recent decades, accurate burned area mapping remains a major challenge due to the complexity of the background and the diversity of the burned areas. This paper presents a novel framework for burned area mapping based on Deep Siamese Morphological Neural Network (DSMNN-Net) and heterogeneous datasets. The DSMNN-Net framework is based on change detection through proposing a pre/post-fire method that is compatible with heterogeneous remote sensing datasets. The proposed network combines multiscale convolution layers and morphological layers (erosion and dilation) to generate deep features. To evaluate the performance of the method proposed here, two case study areas in Australian forests were selected. The framework used can better detect burned areas compared to other state-of-the-art burned area mapping procedures, with a performance of >98% for overall accuracy index, and a kappa coefficient of >0.9, using multispectral Sentinel-2 and hyperspectral PRISMA image datasets. The analyses of the two datasets illustrate that the DSMNN-Net is sufficiently valid and robust for burned area mapping, and especially for complex areas.

**Keywords:** deep learning; PRISMA; burned area; Sentinel-2; morphological operator; convolutional neural network

## 1. Introduction

As a natural hazard, wildfires represent one of the most important reasons for the evolution of ecosystems in the Earth's system on a global scale [1–3]. Recently, the frequency of occurrence of wildfires has increased significantly due to climate change and human activities around the world [4,5]. Wildfires can be influenced by the environment from different aspects, such as soil erosion, increasing flood risk, and habitat degradation for wildlife [6,7]. Furthermore, wildfires generate a wide range of pollutants, including greenhouse gases (i.e., methane and carbon dioxide) [8].

Burned area mapping (BAM) can be useful to predict the behavior of a fire, to define the burning biomass, for compensation from insurance companies, and for estimation of greenhouse gases emitted [9,10]. As result, the generation of reliable and accurate burned area maps is necessary for their management and planning in the support of decision

making. BAM by traditional methods (e.g., field surveys) is a major challenge, and these methods have some limitations, such as the wide areas to be covered and the lack of direct access to the region of interest, which leads to large time and financial costs [10].

The Earth observation satellite fleet has steadily grown over the last few decades [11]. The diversity of Earth observation datasets means that remote sensing (RS) is now known as a key tool in the provision of valuable information about the Earth that is available at low cost and time needs on a global scale [12]. Currently, the upcoming new series of RS sensors (e.g., Landsat-9, *PRecursore IperSpettrale della Missione Applicativa* (PRISMA), Sentinel-5) provides improvements in terms of spatial, temporal, and spectral detail, with RS now becoming a routine tool with an extensive range of applications [13,14]. The most common applications of RS include classification [15,16] and detection of targets [17,18] and changes [19,20].

The diversity of RS Earth observation imagery and its free availability has meant that monitoring of changes following disasters has turned into a hot topic for research [21–28]. Indeed, we are witnessing many BAM products on a global scale that differ in terms of spatial resolution and reliability of the burned areas mapped. Based on spatial resolution, the recent BAM methods can be categorized into two main groups: (1) coarse spatial resolution satellite sensors and (2) fine spatial resolution sensors.

Burned area mapping based on the low and medium resolution of satellite imagery is common in the RS community. In recent years, many studies have used BAM based on Moderate Resolution Imaging Spectroradiometer (MODIS), Sentinel-3, Medium Resolution Imaging Spectrometer (MERIS), and Visible/Infrared Imager Radiometer Suite (VIIRS) [29–31]. However, while these sensors have a high temporal resolution, they suffer from low spatial resolution. Accurate BAM for small areas is a major challenge due to the mixing of pixels. Furthermore, the complex diversity of scenes can result in spectrum gains in one burned pixel to be mixed with some other material. Furthermore, these are based on ruleset classification and manual feature extraction such that the extraction of suitable features and the finding of optimum threshold values are time consuming.

Recently, with the arrival of a new series of cloud computing platforms (e.g., Google Earth Engine, Microsoft Azure), BAM using fine-resolution datasets has been considered by researchers. The capacity of cloud computing platforms has created a great opportunity for BAM based on high-resolution datasets and advanced machine-learning-based methods for accurate mapping. Based on the structure of the algorithm, we can categorize these methods into two main categories: (1) BAM by conventional machine-learning methods and (2) BAM via deep-learning-based frameworks.

Burned area mapping based on conventional machine-learning-based methods can be used to extract spectral and spatial features, and then to define the burned areas according to a classifier [10]. For instance, Donezar, et al. [32] designed a BAM framework based on the multitemporal change-detection method and time series synthetic aperture radar (SAR) imagery. They used an object-based image analysis method for classification of the SAR imagery. They also used the Shuttle Radar Topography Mission (SRTM) for digital elevation models to enhance their BAM results. Additionally, Xulu, et al. [33] considered a BAM method based on differenced normalized burned ratios and Sentinel-2 imagery in the cloud-based Google Earth engine. A random forest classifier method was used for the BAM. They reported an overall accuracy close to 97% for detection of burned areas. Moreover, Seydi, Akhoondzadeh, Amani and Mahdavi [10] evaluated the performance of a statistical machine-learning method for BAM using the Google Earth Engine and pre/post-fire Sentinel-2 imagery. Furthermore, they evaluated the potential spectral and spatial texture features using a Harris hawks optimization algorithm for the BAM. They reported an accuracy of 92% by the random forest classifier on the validation dataset. Liu, et al. [34] proposed a new index for BAM for bi-temporal Landsat-8 imagery and an automatic thresholding method. They evaluated the efficiency of their proposed method in different areas. Their BAM results showed that their presented method had high efficiency.

Recently, deep-learning-based approaches have been applied increasingly for mapping RS imagery, with promising results obtained. These methods can extract high-level features from the raw data automatically, by convolution layers. This advantage of deep-learning-based methods has resulted in their use for BAM. BAM based on deep-learning-based methods has become a hot topic of research, with many methods being proposed. For instance, Nolde, et al. [35] designed large-scale burned area monitoring in near-real-time based on the morphological active contour approach. This framework was applied through several steps: (1) generation of a normalized difference vegetation index (NDVI) for pre/post-fire; (2) determination of the region of interest based on active fires, anomaly detection, and region-growing methodologies; (3) accurate shape of the burned area perimeter extraction based on morphological snakes; (4) confidence evaluation based on a burn area index; and (5) tracking. The result was accuracy of 76% by evaluation with reference data. Knopp, et al. [36] carried out BAM by deep-learning-based semantic segmentation based on mono-temporal Sentinel-2 imagery. They used the U-Net architecture for their BAM. A binary change map was obtained based on the thresholding of the U-Net probability map. They reported the difficulty of the segmentation model in some areas, such as agriculture fields, rocky coastlines, and lake shores. de Bem, et al. [37] investigated the effects of patch sizes on the result of burned area classification with deep-learning methods using Landsat-8 OLI. Here, three different deep-learning methods were investigated: simple convolutional neural network (CNN), U-Net, and Res-U-Net. Their results showed that Res-U-Net had high efficiency, with a patch-size of $256 \times 266$. Hu, Ban and Nascetti [26] evaluated the potential of deep learning methods for BAM based on the unitemporal multispectral Sentinel-2 and Landsat-8 datasets. Their study showed that deep-learning methods have a high potential for BAM in comparison to machine-learning methods. Ban, et al. [38] experimented with the capacity of time series SAR imagery for BAM by a deep-learning method. To this end, their deep-learning framework was based on CNN and was developed to automatically detect burned areas by investigating backscatter variations in the time series of Sentinel-1 SAR imagery. They reported accuracy of <95% for BAM. Zhang, et al. [39] proposed a deep-learning framework for mapping burned areas based on fusion Sentinel-1 and Sentinel-2 imagery. Furthermore, they investigated two scenarios for training the deep-learning method: (1) continuous joint training with all historical data and (2) learning-without-forgetting based on new incoming data alone. They reported that the second scenario for BAM showed accuracy close to 90%, in terms of overall accuracy. Zhang, Ban and Nascetti [39] presented a deep-learning-based BAM framework by fusion of optical and radar datasets. They proposed a deep-learning framework based on CNN, with two convolution layers, max-pooling, and two fully connected layers. They showed an increase in the complexity of the network that resulted in rising computing needs, while the results for the burned area detection were not enhanced. Farasin, et al. [40] presented an automatic framework for evaluation of the damage severity level based on a supervised deep-learning method and post-fire Sentinel-2 satellite imagery. They used double-step U-Net architecture for two tasks (classification and regression). The classification generated binary damage maps and the regression was used to generate damage severity levels. Lestari, et al. [41] increased the efficiency of statistical machine-learning methods and a CNN classifier for BAM using optical and SAR imagery. Their BAM results showed that the CNN method has high efficiency in comparison with other machine-learning methods with texture features. Furthermore, the fusion of optical and SAR imagery can enhance the results of BAM. Belenguer-Plomer, et al. [42] developed a CNN-based BAM method by combining active and passive datasets. Sentinel-1 and Sentinel-2 were used by the CNN algorithm to generate burned areas. Their proposed CNN architecture included two convolution layers, a max-pooling layer, and two fully connected layers. The results of BAM have shown that combining Sentinel-1 and Sentinel-2 imagery can provide improvements.

Although many research efforts have proposed several algorithms for BAM and applied them to fine-resolution optical and SAR RS imagery, many limitations remain: (1) Semantic segmentation based methods (e.g., U-Net DeeplabV3+ and Seg-Net) have

provided promising results, but they need large numbers of labeled datasets, and finding large amounts of sample data with specific sizes (i.e., $512 \times 512$ or $1024 \times 1024$) for small areas is a major challenge. (2) The performance of statistical classification methods such as random forest or support vector machine classifiers depend on the setting of the input features, while the selection and extraction of the informative manual features can be a time-consuming process. (3) Some studies have focused on only spectral features for BAM, while the efficiency of spatial features in BAM has been shown in many studies; furthermore, an unsupervised thresholding manner on the spectral index is not always effective due to the complexity of the background and ecosystem characteristics, and to the topographic effects on the surface reflectance [43,44]. (4) Shallow feature representation methods have been shown not to be applicable in complex areas, especially for BAM tasks. (5) More methods have focused on time series Sentinel-1 imagery; however, preprocessing and processing of SAR imagery is very difficult due to noise conditions.

To overcome these problems, the present study presents a novel framework for BAM with heterogeneous datasets that has many advantages compared to other state-of-the-art methods. The method proposed here is applied in an end-to-end manner without additional processes, based on a deep morphological network. This method is based on change detection that uses pre/post-fire datasets based on deep Siamese morphological operators. Additionally, the efficiency of the hyperspectral dataset in comparison with the multispectral dataset shows that this study takes advantage of the hyperspectral dataset. The proposed framework is additive with the type of datasets, whereby the pre-event dataset is Sentinel-2 imagery while the post-event dataset can be either Sentinel-2 or hyperspectral PRISMA datasets.

The main contributions of this study follow: (1) BAM is based on deep morphological layers for the first time; (2) it takes advantage of the hyperspectral PRISMA sensor dataset for accurate BAM for the first time; and (3) it includes evaluation of the performance of the multispectral and hyperspectral dataset in BAM and comparison of the results with state-of-the-art methods.

This paper is outlined as follows: Section 2 provides the details of the DSMNN-Net for BAM. Section 3 introduces the study areas and the datasets. The evaluation results of this study area are provided in Section 4, and the experimentation results are discussed in Section 5.

## 2. Methodology

The proposed framework is conducted in three steps, according to the flowchart in Figure 1. The first step is image preparation, and in this step, some preprocessing (i.e., registration) is applied. The second step is the training of the proposed network to tune the network parameters based on reference sample data. The training and validation datasets are exploited in the training process to optimize the model parameters, while the testing dataset is used to evaluate model hyperparameters. The third step is burned area map generation and accuracy assessment of the result of the BAM.

### 2.1. Proposed Deep Learning Architecture

The proposed DSMNN-Net architecture for the detection of burned areas is illustrated in Figure 2. Accordingly, the framework has two parts: (1) two streams of deep-feature-extraction models and (2) classification. The deep-feature-extraction task is conducted in a double-stream manner, such that these streams are for post-event and pre-event datasets, respectively. Then, the deep features are transformed to the next task, which is the classification. The classification task included two fully connected layers and a soft-max layer for making a decision. More details of the DSMNN-Net are explained in the next subsection.

### 2.2. Deep-Feature Extraction

Feature extraction can be defined as an image processing technique to determine the identity of the mutual importance of imaged areas. There are many procedures for feature extraction in the field of image processing and RS, such as the texture Haralick feature [45,46], spectral features (e.g., the NDVI) [10], and transformation-based (e.g., principal component analysis) and deep-feature [15,47,48] extraction. Among the types of feature-extraction methods, the deep-feature-extraction methods have found a specific place in RS communication because they have great potential for the extraction of complex features from an image [49]. Deep-learning methods can automatically extract high-level spatial and spectral features simultaneously [50]. This advantage of deep-learning methods means that they have been used for many applications in RS, such as change detection [51], classification [52], anomaly detection [53], and damage mapping [54].

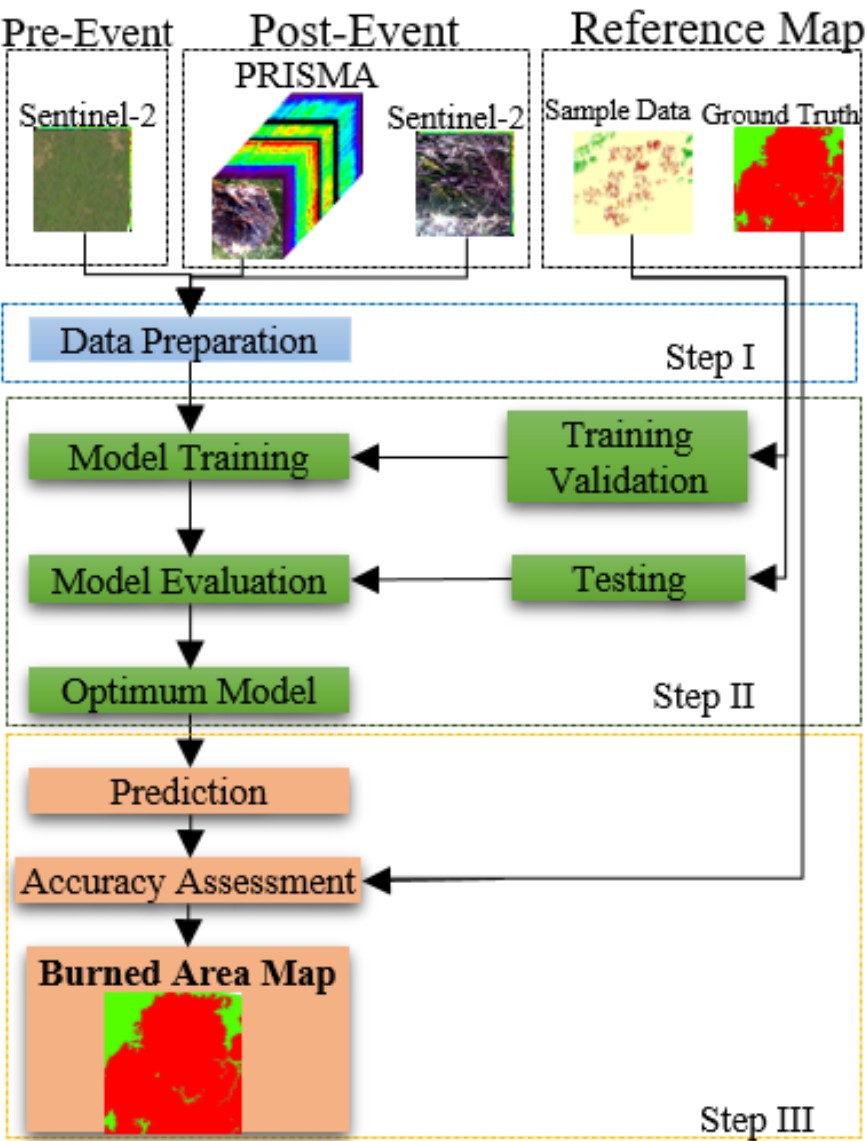

**Figure 1.** Overview of the general framework for the burned area mapping.

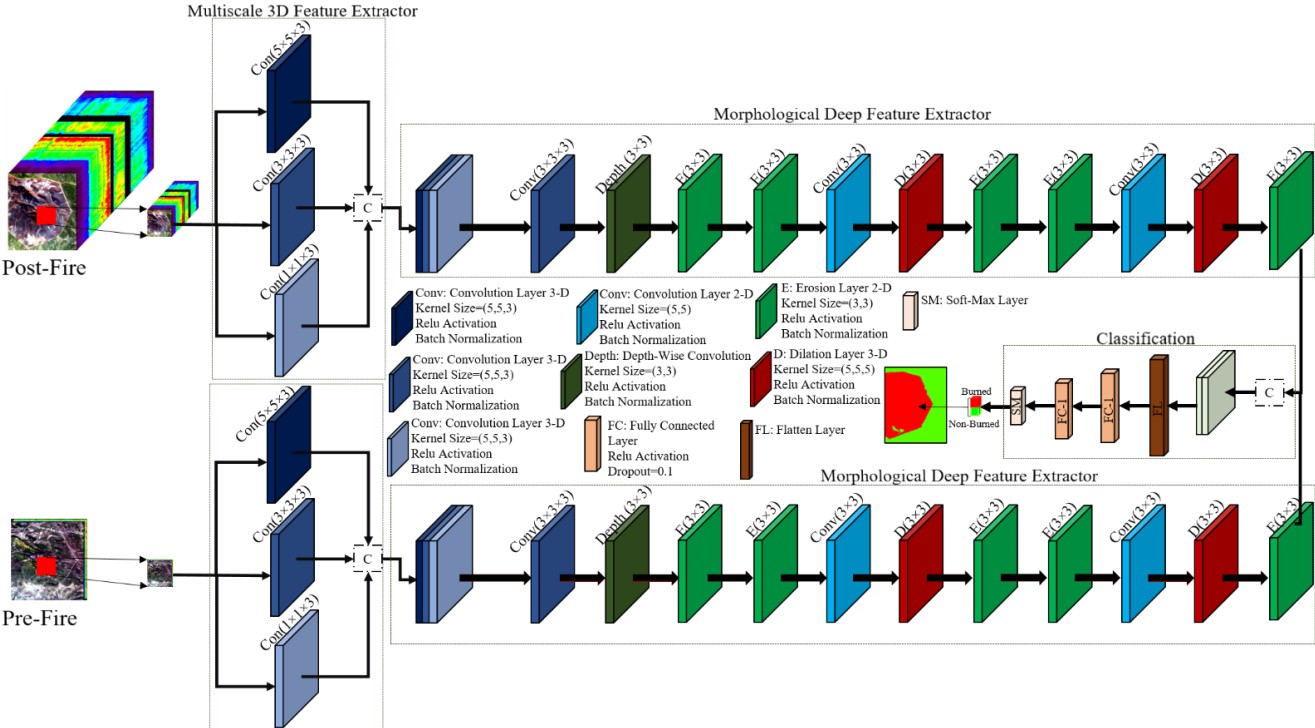

**Figure 2.** Overview of the proposed DSMNN-Net architecture for burned area mapping.

The deep features are extracted by convolution layers, and the arrangement of the convolution layers and their diversity has caused many deep-learning-based methods to be proposed [55–57]. Presenting the informative structure of convolution layers can be a major challenge. In this regard, the present study presents a novel framework based on standard 3D, 2D, and depthwise convolution layers, with their combination with morphological layers. As illustrated in Figure 2, the method proposed here has two deep-feature extractor streams. The first stream investigates the pre-fire dataset, and the second stream explores the deep features from the post-event dataset. Each stream includes 2D-depthwise, 3D/2D standard convolution layers, and morphological layers based on erosion and dilation operators. Initially, the deep-features extraction is based on 3D multiscale convolution layers, and then the extracted features are fed into the 3D convolution layer. The main advantage of 3D convolution layers is to take the full content of the spectral information of the input dataset by considering the relation among all of the spectral bands. Furthermore, the multiscale block enhances the robustness of the DSMNN-Net against variations in the object size [12]. The multiscale block uses a type kernel size of convolution layers that increase the efficiency of the network. The expected features are reshaped and converted to 2D feature maps, and then the 2D-depthwise convolution layers are used. Next, the hybrid morphological layers based on 2D dilation and erosion combine with 2D convolution layers to explore more high-level features. For this, first, we use two erosion layers, and then the 2D convolution layer and dilation layers are used (see Figure 2). Finally, the 2D convolution, erosion, and dilation layers have been used in the last part of the morphological deep-feature extractor. The extracted deep features are concatenated for two streams and then they are flattened and transferred to two fully connected layers, and finally, the soft-max layer is entitled to decide the input data. The main differences between the proposed architecture and other CNN frameworks are:

(1)  We take advantage of multiscale convolution layers that increase the robustness of the network against the scale of variations.

(2)  We use the trainable morphological layers, which can increase the efficiency of the network for the extraction of nonlinear features.

(3)   We use 3D convolution layers to make use of the full content of the spectral information in the hyperspectral and multispectral datasets.

(4)   We use depthwise convolution layers that are computationally cheaper and can help to reduce the number of parameters and to prevent overfitting.

### 2.3. Convolution Layer

The convolution layers are the core building block of deep-learning methods that can learn feature representations of input data. A convolution layer builds several convolution kernels to extract the type of meaningful features. This study used 3D/2D convolution layers for deep-feature extraction [58–60]. Mathematically, the feature value ($\Psi$) in the $l$th layer is expressed according to Equation (1) [61]:

$$\nu^l = g\left(w^l x^{l-1}\right) + b^l,\tag{1}$$

where $x$ is the input data, $g$ is the activation function, $b$ is the bias vector for the current layer, and $w$ is the weighted vector. The value ($\nu$) at position ($x,y,z$) on the $j$th feature $i$th layer for the 3D convolution layer is given by Equation (2) [62]:

$$\nu_{i,j}^{xyz} = g(b_{i,j} + \sum_{\chi} \sum_{\omega=0}^{\Omega_i-1} \sum_{\varphi=0}^{\Phi_i-1} \sum_{\lambda=0}^{\Lambda_i-1} W_{i,j,\chi}^{\omega,\varphi,\lambda} v_{i-1,\chi}^{(x+\omega)(y+\varphi)(z+\lambda)})\tag{2}$$

where $\chi$ is the feature cube connected to the current feature cube in the ($i-1$)th layer, and $\Omega$, $\Phi$, and $\Lambda$ are the length, width, and depth of the convolution kernel size, respectively. In 2D convolution, the output of the $j$th feature map in the $i$th layer at the spatial location of ($x,y$) can be computed using Equation (3):

$$\nu_{i,j}^{xy} = g\left(b_{i,j} + \sum_{\chi} \sum_{\omega=0}^{\Omega_i-1} \sum_{\varphi=0}^{\Phi_i-1} W_{i,j,\chi}^{\omega,\varphi} v_{i-1,\chi}^{(x+\omega)(y+\varphi)}\right)\tag{3}$$

### 2.4. Morphological Operation Layers

Topological operators are applied to images by morphological operators to recover or filter out specific structures [63,64]. Mathematical morphology operators are nonlinear image operators that are based on the image spatial structure [65–67]. *Dilation* and *Erosion* are shape-sensitive operations that can be relatively helpful to extract discriminative spatial-contextual information during the training stage [67–69]. *Erosion*($\ominus$) and *Dilation*($\oplus$) are two basic operations in morphology operators that can be defined for a grayscale image $X$ with size $M \times N$ and $W$ structuring elements, as follows in Equation (4) [65,66]:

$$\begin{aligned}(X \oplus W)(x,y) = \max_{(i,m)\in S}(X(x-l,y-m) + W_d(l,m)) \\ S = \{(l,m)|l \in \{1,2,3,\dots,a\}; m \in \{1,2,3,\dots,b\};\}\end{aligned}\tag{4}$$

where $W_d$ is the structuring element of dilation that can be defined on domain $S$. Accordingly, the erosion operator with structuring element $W_d$ can be defined as follows in Equation (5):

$$(X \ominus W)(x,y) = \min_{(i,m)\in S}(X(x+l,y+m) - W_e(l,m))\tag{5}$$

The structure element is initialized based on random values in the training process. The back-propagation algorithm is used to update the structure elements in the morphological layers. The propagation of the gradient through the network is very similar to that of a neural network.

### 2.5. Classification

After deep-feature extraction by convolution and morphological layers, the deep features are transformed for the flattening layer to reshape as 1D vectors. Then, these vectors are fed to the first fully connected layer and the second fully connected layer. The latest layer is soft-max, which assigns probabilities to each class for input pixels. Figure 1 presents the classification procedure for this framework.

*2.6. Training Process*

The network parameters are initialized based on the initial values and then are tuned iteratively based on optimizers, such as stochastic gradient descent. The DSMNN-Net is trained based on the training data, and the error of the network is obtained based on the calculation of the loss value on the validation dataset. The error of the training model is fed to the optimizer and is used to update the parameters. Due to back-propagation, the parameters are updated at each step to decrease the error of comparing the results obtained from the network with the validation dataset. The Tversky loss function is used to calculate the network error in the training process, which is a generalization of the dice score [70]. The Tversky index (*TI*) between $\hat{\Psi}$ (predicted value) and $\Psi$ (truth value) is defined as in Equation (6):

$$TI(\hat{\Psi}, \Psi, \alpha, \beta) = \frac{|\hat{\Psi}\Psi|}{|\hat{\Psi}\Psi| + \alpha|\hat{\Psi}/\Psi| + \beta|\Psi/\hat{\Psi}|} \tag{6}$$

where $\alpha$ and $\beta$ control the magnitude of penalties for false positive and false negative pixels, respectively. These parameters are often chosen based on trial and error.

*2.7. Accuracy Assessment*

We assessed the results of the BAM based on visual and numerical analysis. The numerical analysis was applied as the standard measurement indices. To this end, the five most common quantitative assessment metrics were selected to evaluate the results. These indices are the overall accuracy (OA), the kappa coefficient (KC), and the F1-score, Recall, and intersection over union (IOU).

To compare the performance of the method proposed here, two state-of-the-art deep-learning methods were selected for this study. The first method was the deep Siamese network, which has been proposed in many studies for change detection purposes [71–73]. This method has three convolution layers in each stream, and then fully connected was used for classification. Then, the second method was CNN, based on a framework designed by Belenguer-Plomer, Tanase, Chuvieco and Bovolo [42] for mapping of burned areas. This method has two convolution layers and a max-pooling layer, then two fully connected layers were used. More details of this method can be found in [42].

## 3. Case Study and Satellite Images

This section investigates the case study area and the satellite data in more detail.

*3.1. Study Area*

Both study areas in this research were located in the Australian continent. The main reason for choosing the areas was the availability of the PRISMA hyperspectral datasets for these areas. Reference is the most important factor in the evaluation of BAM results. Thus, the reference data were obtained based on visual analysis and the interpretation of the results of BAM in previous papers. Figure 3 presents the locations of two study areas, in the southern Australian continent.

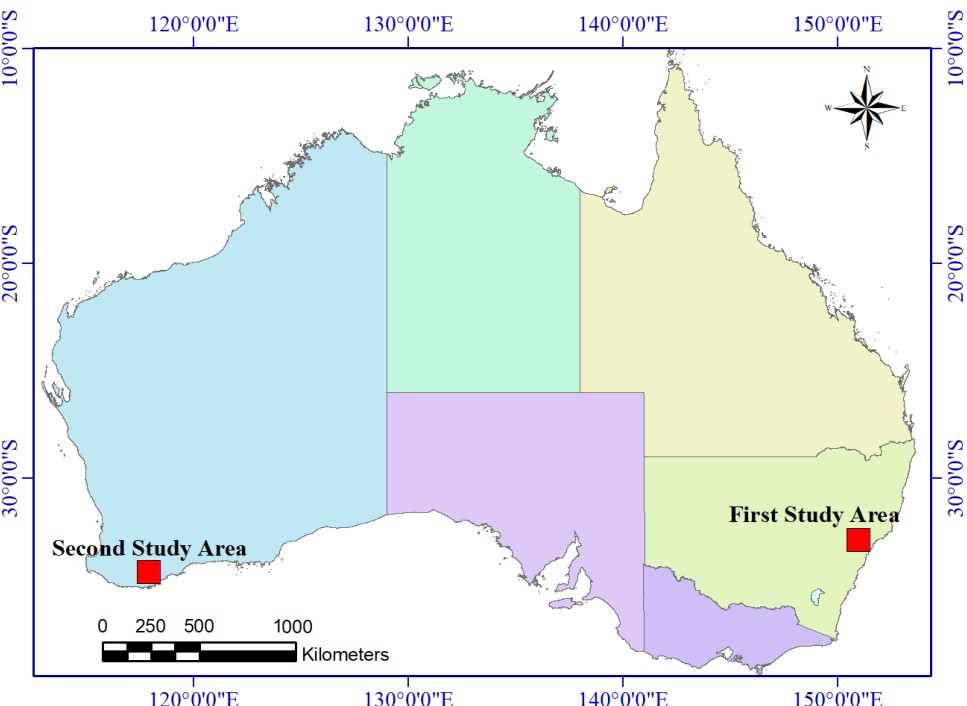

**Figure 3.** The locations of the two study areas for burned area mapping.

Figure 4 shows the incorporated burned area datasets for the first study area. Figure 5 illustrates the original incorporated dataset for the BAM for the second study area. The details for the incorporated datasets for both of the study areas are given in Table 1.

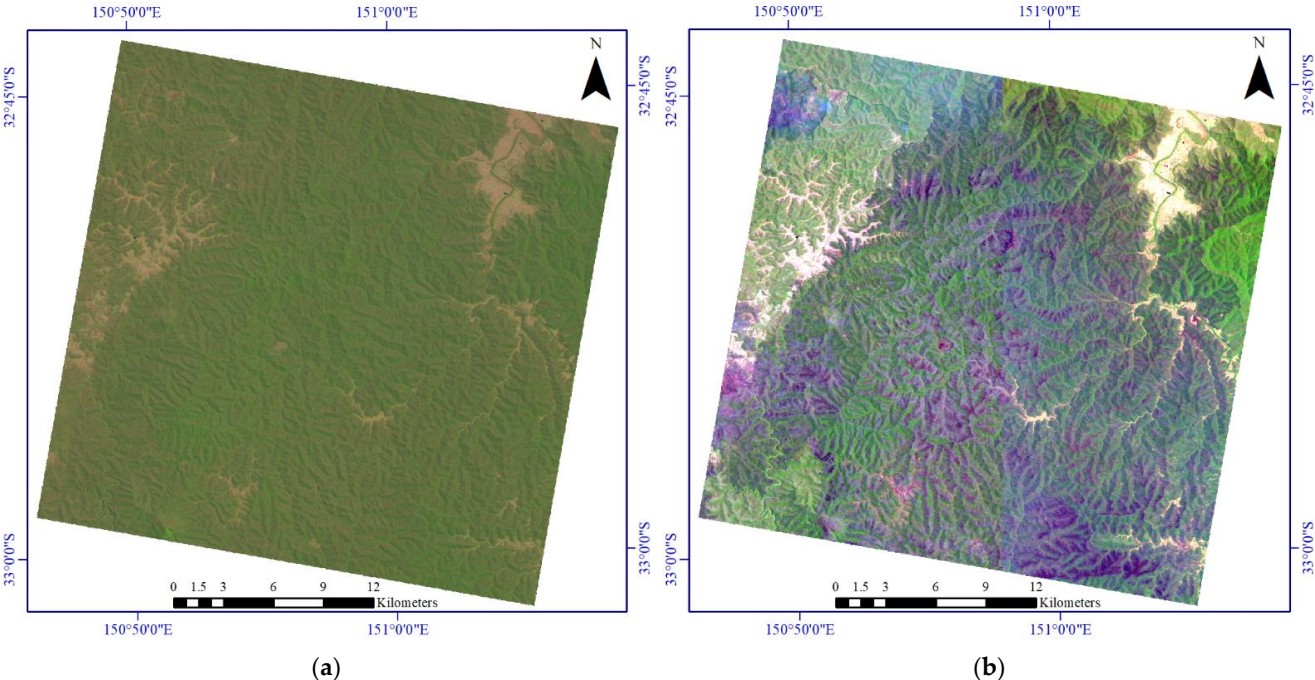

(**a**)  (**b**)

**Figure 4.** *Cont.*

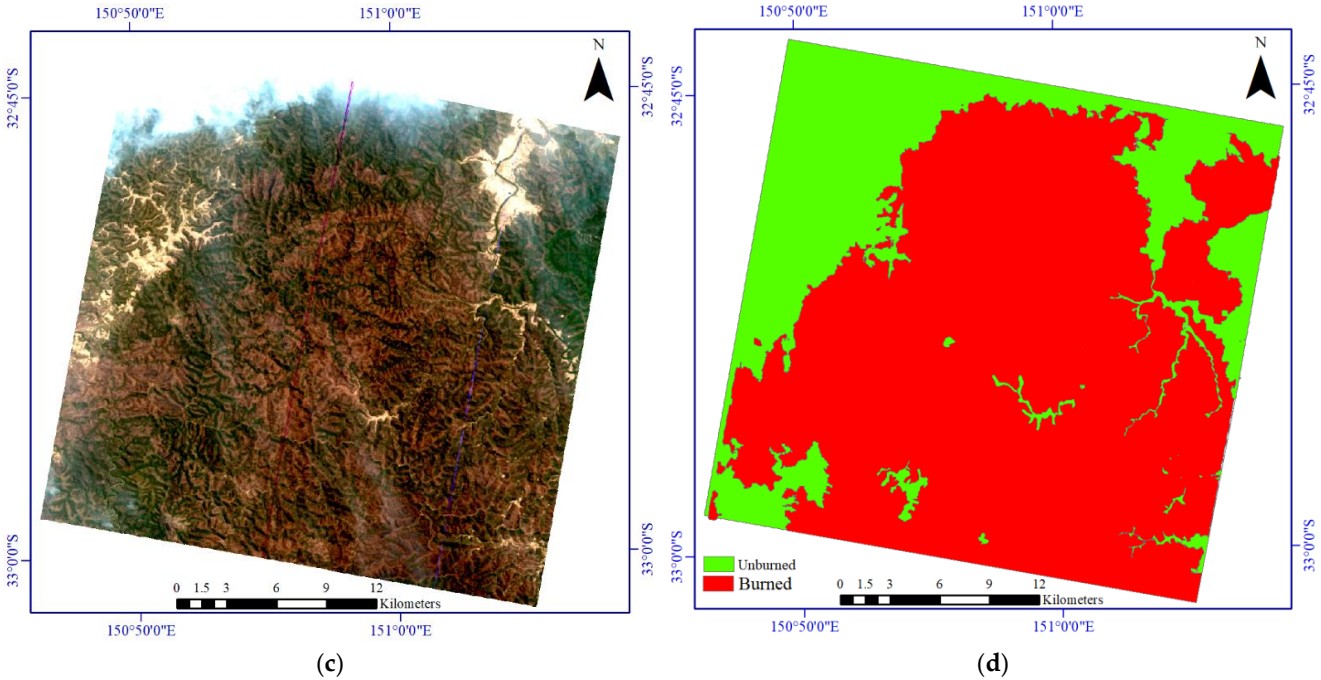

**Figure 4.** The dataset used for the burned area mapping for the first study area. (**a**) Pre-event Sentinel-2 imagery. (**b**) Post-event Sentinel-2 dataset. (**c**) Post-event PRISMA hyperspectral imagery. (**d**) Ground truth.

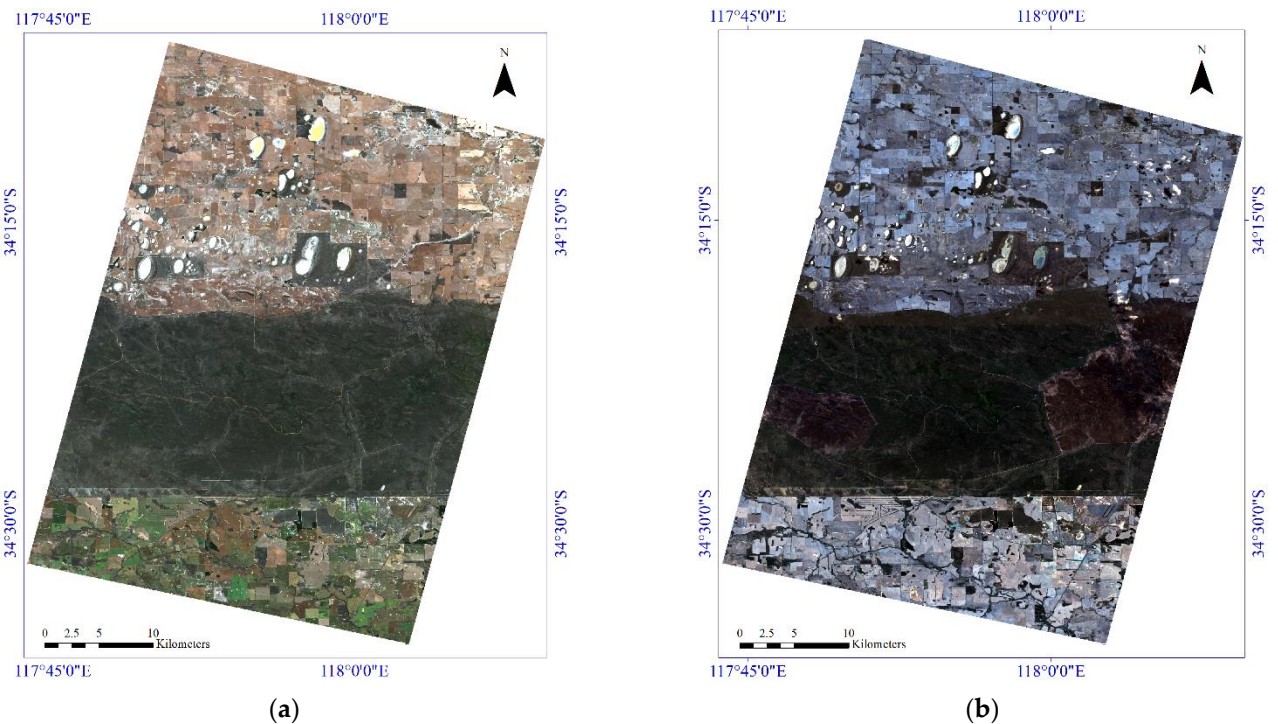

**Figure 5.** *Cont.*

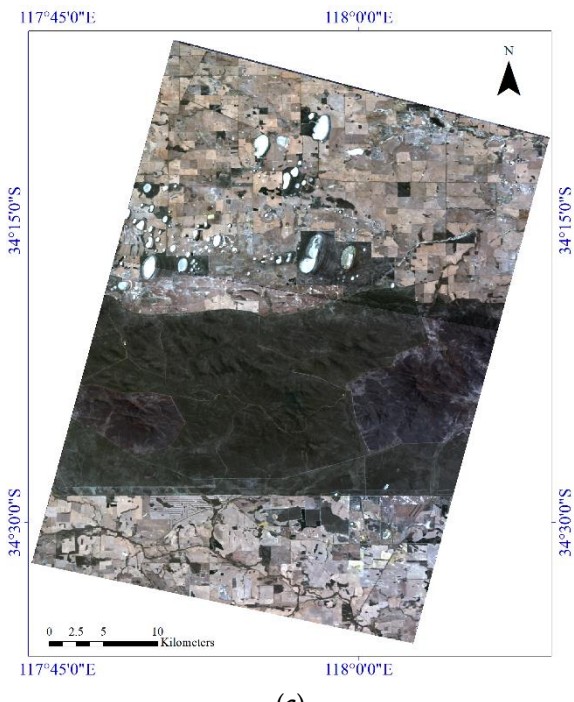

(**c**)

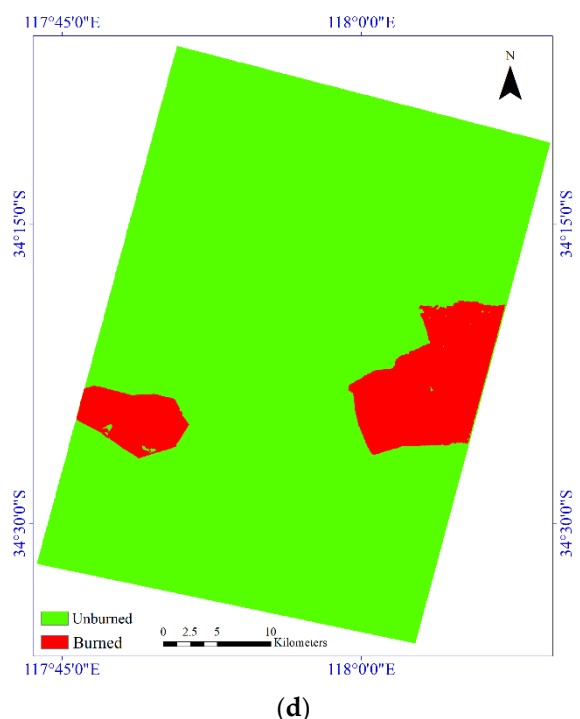

(**d**)

**Figure 5.** Illustration of the various incorporated datasets for the burned area mapping for the second study area. (**a**) Pre-event Sentinel-2 imagery. (**b**) Post-event Sentinel-2 dataset. (**c**) Post-event PRISMA hyperspectral imagery. (**d**) Ground truth.

**Table 1.** The main characteristics of the incorporated datasets for both case studies.

| Sensor | Properties | First Study Area | Second Study Area |
|---|---|---|---|
| Sentinel-2 | Spectral bands | 13 | 13 |
| | Spatial resolution (m) | 10 | 10 |
| | Resampled spatial resolution (m) | 30 | 30 |
| | Data size (pixel) | 1168 × 1168 | 1159 × 1853 |
| | Pre-event acquired date | December 2019 | October 2019 |
| | Post-event acquired date | November 2020 | January 2020 |
| PRISMA | Spectral bands | 169 | 169 |
| | Spatial resolution (m) | 30 | 30 |
| | Data size (pixel) | 1168 × 1168 | 1159 × 1853 |
| | Post-event acquired date | December 2019 | January 2020 |

### 3.2. Sentinel-2 Images

Sentinel-2 is a European Space Agency Earth observation project that provides continuity to services dependent on multispectral high-spatial-resolution observations over the whole land surface of the Earth. This mission consists of two satellites, Sentinel-2-A and Sentinel-2-B, which have completed the existing Landsat and Spot missions and have enhanced data availability for RS communications. One satellite has a temporal resolution of 10 days, while the two satellites have a temporal resolution of 5 days [10]. The Sentinel-2 main sensor, the multispectral instrument, is based on the push-broom principle. Sentinel-2 has 13 spectral bands and broad spectral coverage.

This study used the Level-2A product as input data for BAM, which are surface reflectance data. Furthermore, it was necessary to convert the spatial resolution of the Sentinel-2 dataset into the spatial resolution of the hyperspectral dataset (30 m).

### 3.3. PRISMA Images

PRISMA is a medium-resolution hyperspectral imaging mission of the Italian Space Agency that was launched in March 2019 [74]. The PRISMA sensor is a spaceborne system that acquires hyperspectral datasets continuously, with a repeat orbital cycle of approximately 29 days [75]. PRISMA images the Earth surface in 240 contiguous spectral bands (66 visible to near-infrared, plus 174 short-wave infrared) with a push-broom scanning mode, covering the wavelengths between 400 and 2500 nm, at a spatial resolution of 30 m. The high dimensional spectral bands provide the possibility to analyze complex land-cover objects [14]. We chose the level-2-D product for the BAM, which was preprocessed (i.e., atmospheric correction and geolocation, orthorectification) [14]. The PRISMA hyperspectral dataset is freely available on this website: http://prisma.asi.it/missionselect/, accessed on 16 November 2021. After removing the noisy and no-data bands, 169 spectral bands were chosen for the next analysis.

### 4. Experiments and Results

Gathering of sample data is required to estimate the burned area due to using a supervised learning method. The quality and quantity of the sample data have a key role in BAM. In this study, the numbers of the sample data were kept at a sufficient level for the two classes (i.e., burned areas, unburned areas). Figure 6 shows the spatial distribution of the sample data.

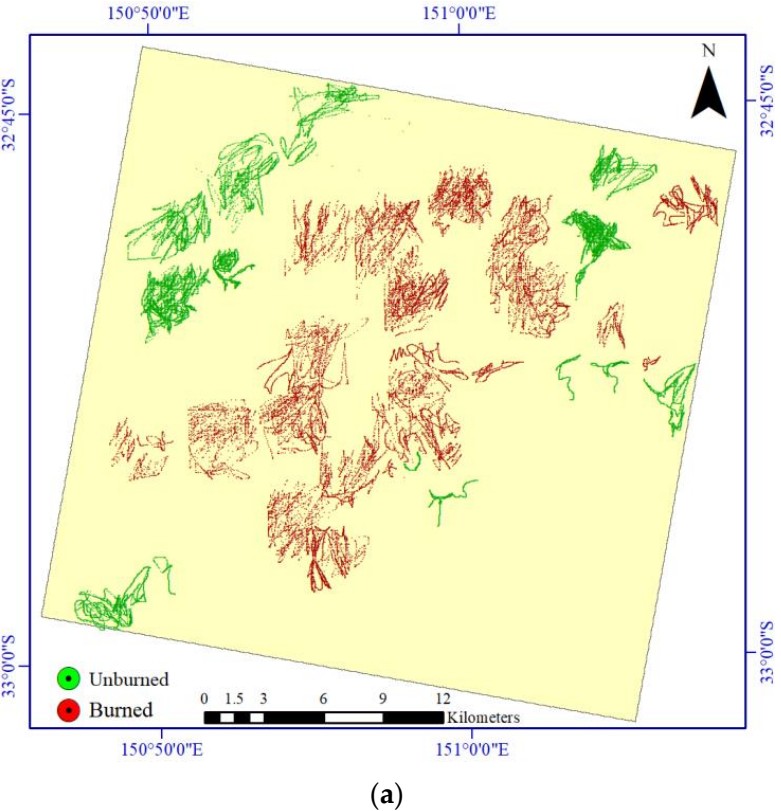

(**a**)

**Figure 6.** *Cont.*

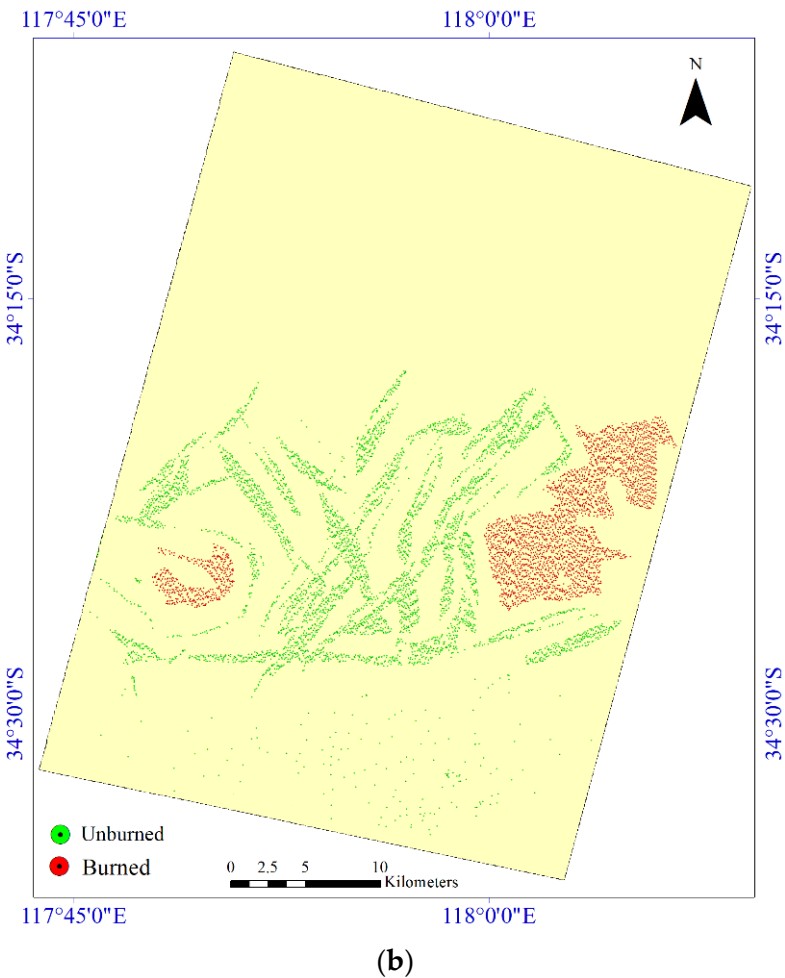

**(b)**

**Figure 6.** The spatial distributions of the two study areas for the burned area mapping. (**a**) Sample data for the first study area. (**b**) Sample data for the second study area.

In addition, Table 2 shows the sizes of sample data for two classes in the study areas.

**Table 2.** The number of samples used for mapping of burned area in the two study areas.

| Case Study | Number of Pixels in the Study Area | Class | Number of Samples | Training | Validation | Testing |
|---|---|---|---|---|---|---|
| First study area | 989,764 | Unburned<br>Burned | 15,318<br>21,387 | 9803<br>13,687 | 2450<br>3421 | 3065<br>4459 |
| Second study area | 1,955,898 | Unburned<br>Burned | 6590<br>3206 | 4217<br>2051 | 1054<br>513 | 1318<br>642 |

### 4.1. Parameter Setting

The DSMNN-Net has hyperparameters that need to be set. These hyperparameters were set manually based on trial and error. The optimum values of these parameters were set as follows: the input patch-size for Sentinel-2 and PRISMA sensors were $11 \times 11 \times 13$ and $11 \times 11 \times 169$, respectively, with 500 epochs; the weight initializer was set as He-normal-Initializer [76] for convolution layers; the random value for initializing of the morphological layers, number of neurons at the fully-connected layer was 900; the initial learning rate was $10^{-4}$; and the minibatch size was 550. It is worth noting that all of the hyperparameters were constant during the process for all of the CNN methods. Similarly, the two other methods set such values. Additionally, the selection of some of these

parameters was related to hardware (e.g., increasing minibatch size quickly filled the RAM of the system). Moreover, the weight initializer by the He-normal-initializer increased the speed of network convergence compared to the random initializer.

### 4.2. Results

The results for the BAM for the two study areas are considered in this section. For the two main scenarios, these were investigated according to Table 3.

**Table 3.** Different scenarios for mapping of burned areas in two case-study areas.

| Scenario | Pre-Event Dataset | Post-Event Dataset |
|:---:|:---:|:---:|
| S#1 | Sentinel-2 | Sentinel-2 |
| S#2 | Sentinel-2 | PRISMA |

### 4.2.1. First Study Area

Figure 7 shows the results of the BAM based on the post/pre-event Sentinel-2 imagery. Based on these results, the DSMNN-Net differed from the BAM. Most methods detected the burned areas, with differences seen in the detail. For example, there are some missed detection areas in the results of the two CNN-based methods (center of scene) while the method proposed here detected these well.

Figure 8 shows the mapping results for the heterogeneous dataset provided by various methods. As shown in Figure 8, all of the methods provided better performance in comparison with the first scenario (S#1) as a result of the small missed detection area that was significantly decreased. The results of the DSMNN-Net fit better with ground truth while results of other methods have many false pixels; in particular, for the Siamese network (Figure 8a). The main differences among these results are obvious at the edges of the burned areas.

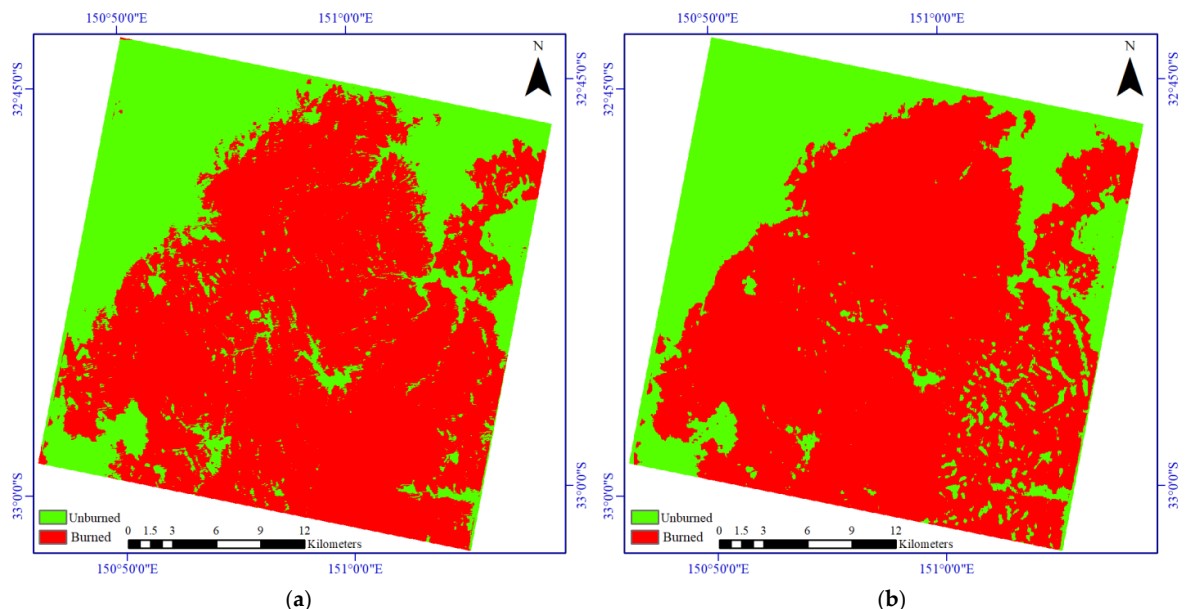

(**a**)　　　　　　(**b**)

**Figure 7.** *Cont.*

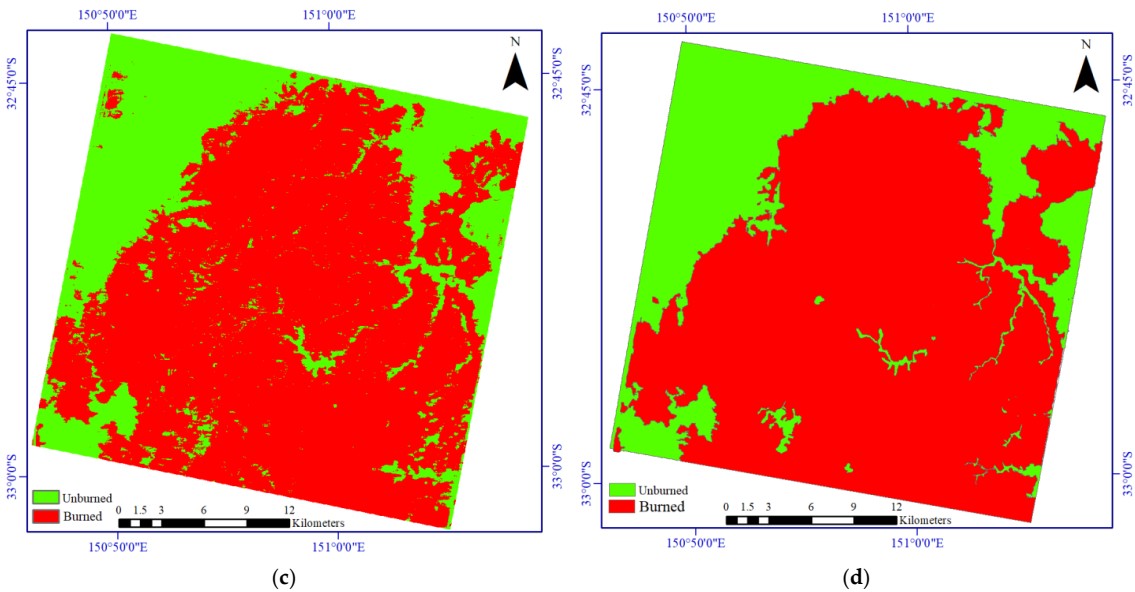

**Figure 7.** Visual comparisons of the results for the burned area mapping based on the post/pre-event Sentinel-2 imagery for the first study area. (**a**) Deep Siamese network. (**b**) Using the CNN method proposed in [42]. (**c**) Using the method proposed in the present study. (**d**) Ground truth map.

The numerical results for the BAM for the first study area are given in Table 4. Based on these data, the accuracies of all of the methods in both scenarios were >87% in all terms. The accuracy of the algorithms in combining the hyperspectral datasets with the multispectral dataset was significantly better than only the multispectral datasets. The accuracy of the BAM results based on the fusion of the PRISMA imagery and Sentinel-2 imagery was >94% by OA index. However, the results of BAM based on only the Sentinel-2 imagery were very close together, but these were considerably different in the second scenario (S#2). The method proposed here provided an accuracy of >97% in terms of the OA, Recall, and F1-score indices. Furthermore, this provided the highest score by KC index for the second scenario.

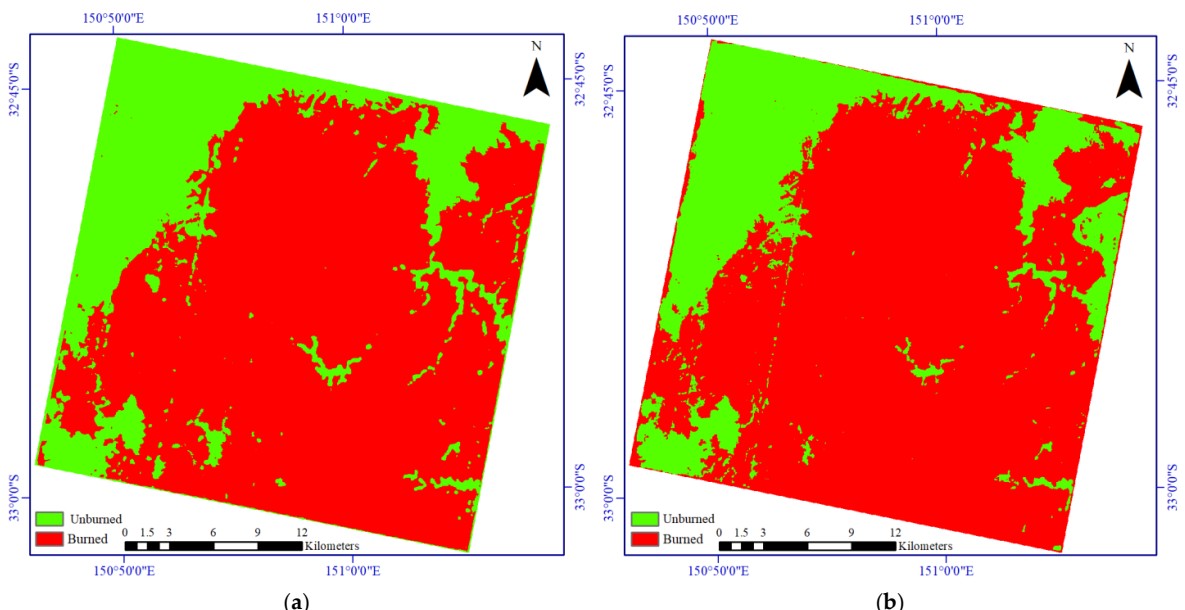

**Figure 8.** *Cont.*

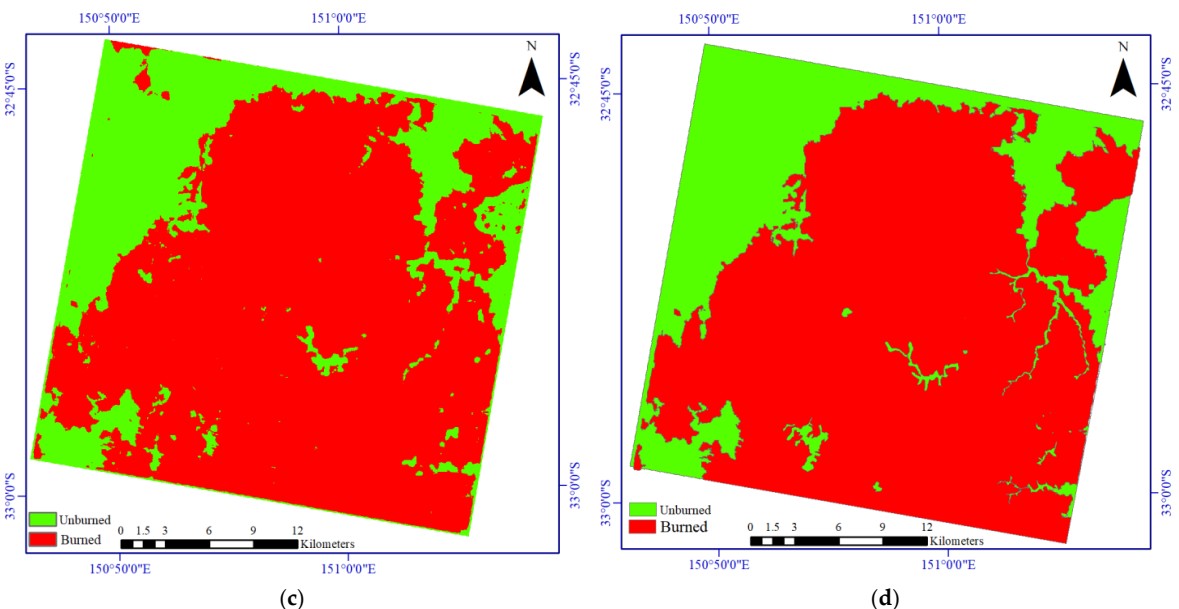

**Figure 8.** Visual comparisons of the results for the burned area mapping based on pre-event Sentinel-2 and post-event PRISMA imagery for the first study area. (**a**) Deep Siamese network. (**b**) Using the CNN method proposed in [42]. (**c**) Using the DSMNN-Net. (**d**) Ground truth map.

**Table 4.** Accuracy assessment for the burned area mapping for the first study area. S#1, pre/post-event Sentinel-2 imagery; S#2, pre-event Sentinel-2 imagery and post-event PRISMA imagery.

| Method | Scenario | OA (%) | Recall (%) | F1-Score (%) | IOU | KC |
|---|---|---|---|---|---|---|
| Siamese network | *S#1* | 87.94 | 87.10 | 91.34 | 0.740 | 0.716 |
| | *S#2* | 94.79 | 96.19 | 96.43 | 0.786 | 0.868 |
| CNN method proposed by [42] | *S#1* | 89.35 | 89.40 | 92.46 | 0.842 | 0.744 |
| | *S#2* | 94.35 | 97.13 | 96.17 | 0.851 | 0.853 |
| DSMNN-Net | *S#1* | **90.24** | **92.51** | **93.26** | **0.864** | **0.755** |
| | *S#2* | **97.46** | **97.99** | **98.25** | **0.901** | **0.936** |

OA, overall accuracy; IOU, intersection over union; KC, kappa coefficient.

### 4.2.2. Second Study Area

Figure 9 illustrates the results of the BAM based on the bi-temporal multispectral/hyperspectral datasets for the second study area. Based on these results, there are some differences among the algorithms seen for the details. Figure 9a shows the performance of the deep Siamese network, in that it has low false pixels, although many missed detection pixels can be seen in the result. However, the lowest missed pixels can be seen in the BAM for the method proposed by [42] in Figure 9b, although it shows high false pixels in the results presented. The result of BAM by the DSMNN-Net can be seen in Figure 9c, which shows the lowest false pixels and missed pixels in the mapping.

The results of the BAM based on the Sentinel-2 and PRISMA sensors for the second area are presented in Figure 10. Based on the comparisons of the results presented with the multispectral dataset, there are some improvements in the details of the mapping. These improvements are more evident in the results of the DSMNN-Net, as some false pixels were classified correctly.

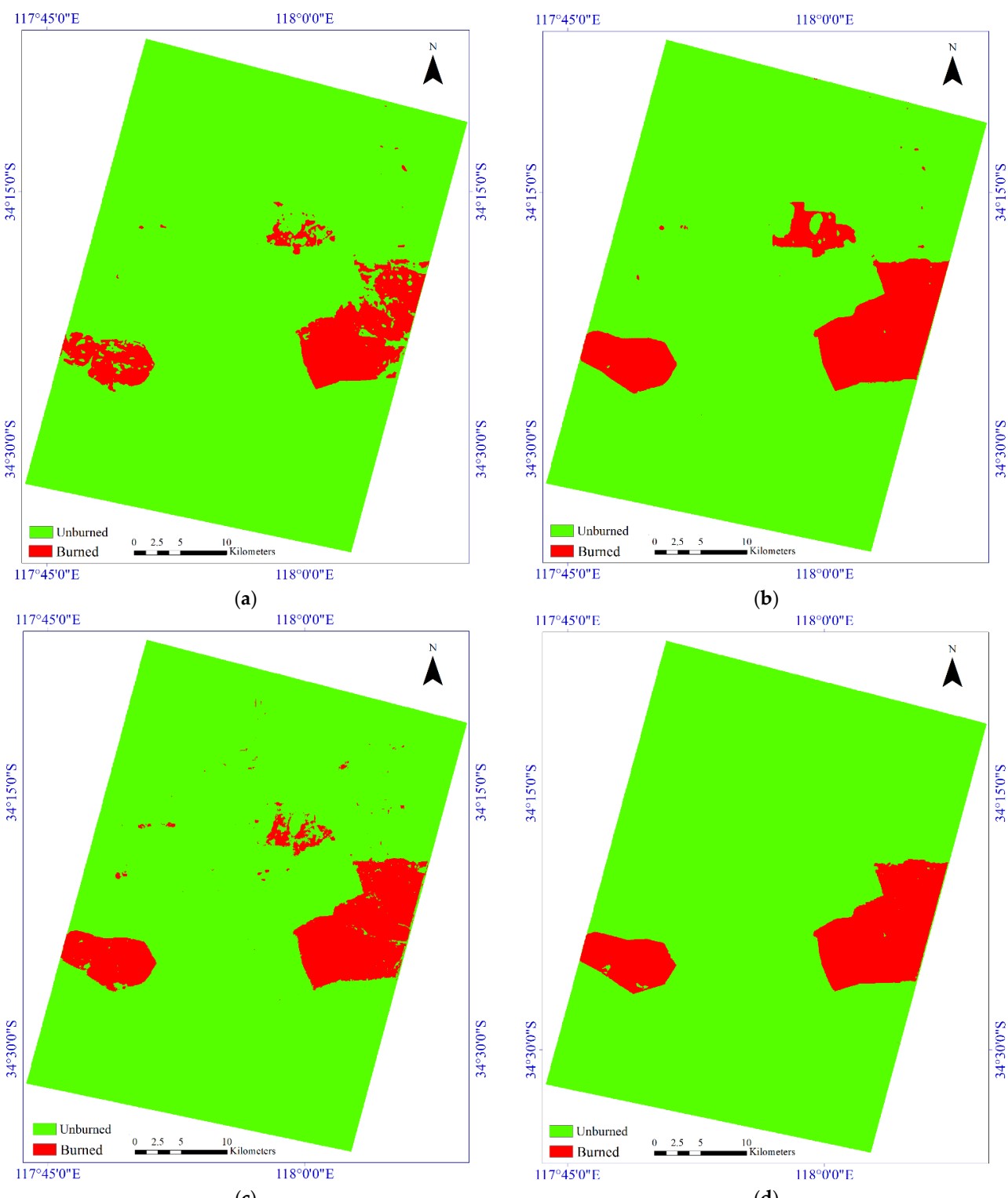

**Figure 9.** Visual comparisons of the results for the burned area mapping based on the post/pre-event Sentinel-2 imagery for the second study area. (**a**) Deep Siamese network. (**b**) Using the CNN method proposed in [42]. (**c**) Using the method proposed in the present study. (**d**) Ground truth map.

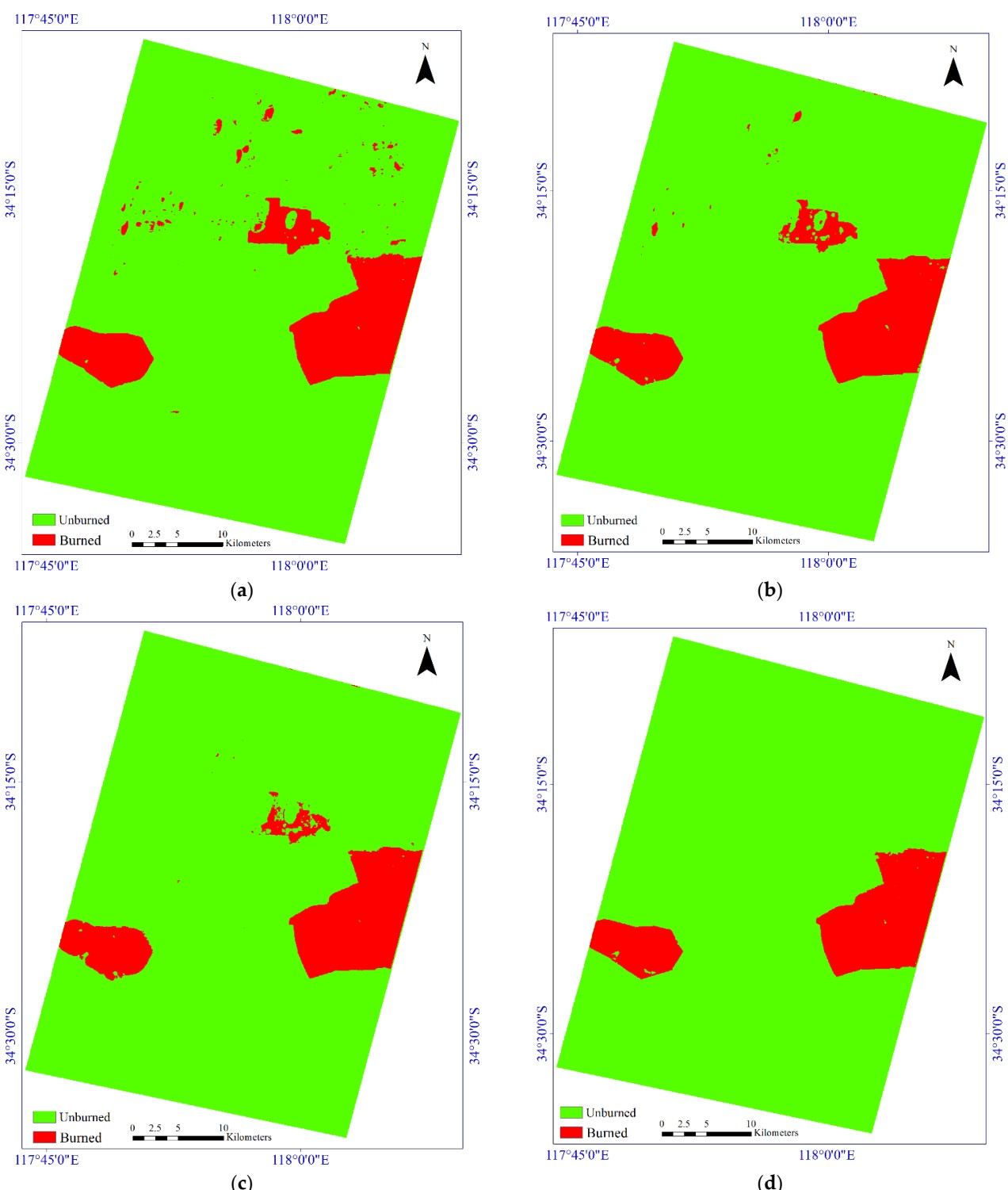

**Figure 10.** Visual comparisons of the results for the burned area mapping based on the pre-event Sentinel-2 and post-event PRISMA imagery for the second study area. (**a**) Deep Siamese network. (**b**) Using the CNN method proposed in [42]. (**c**) Using the DSMNN-Net. (**d**) Ground truth map.

Table 5 presents the quantitative performance comparison of the methods for the second study area for the BAM. Based on these data, there are some enhancements in the results of the BAM in all terms for the case study area. The enhancement of the methods is more evident for the Recall, F1-Score, IOU, and KC indices. For example, the difference between Recall of the Siamese network in the first and second scenarios is >20%; moreover,

in other terms, this improvement is >3%. The improvement of the BAM in the CNN method proposed by [42] was slight. The DSMNN-Net has some improvements that are more evident by terms KC, IOU, F1-Score, and Recall. For example, these terms show greater improvement for the method proposed here in the detection of burned pixels, while there is some improvement in the detection of nonburned pixels.

**Table 5.** Accuracy assessment for the burned area mapping for the second study area. S#1, pre/post-event Sentinel-2 imagery; S#2, pre-event Sentinel-2 imagery and post-event PRISMA imagery.

| Method | Scenario | OA (%) | Recall (%) | F1-Score (%) | IOU | KC |
|---|---|---|---|---|---|---|
| Siamese network | *S#1* | 97.32 | 78.90 | 85.03 | 0.739 | 0.835 |
| | *S#2* | 97.41 | 98.79 | 88.05 | 0.786 | 0.866 |
| CNN method proposed by [42] | *S#1* | 98.21 | 98.94 | 91.44 | 0.842 | 0.904 |
| | *S#2* | 98.35 | 97.75 | 91.94 | 0.851 | 0.910 |
| DSMNN-Net | *S#1* | **98.56** | **95.13** | **92.75** | **0.864** | **0.919** |
| | *S#2* | **98.95** | **98.90** | **94.80** | **0.901** | **0.942** |

OA, overall accuracy; IOU, intersection over union; KC, kappa coefficient.

## 5. Discussion

This study focused on BAM based on deep-learning methods based on bi-temporal multispectral and hyperspectral imagery. BAM has mainly been applied based on low-resolution satellite imagery (e.g., MODIS, VIIRS, and Sentinel-3). However, BAM based on these sensors has provided promising results, although mapping of small burned areas is the most important challenge. These methods support the high coverage areas but do not provide suitable results for small areas. Furthermore, there are some burned area products on a global scale based on the MODIS satellite imagery. Many studies have evaluated the accuracy obtained by BAM based on the MODIS collection, where this has been reported as <80%, while for the BAM for both study areas, the DSMNN-Net provided an accuracy of >98% by the OA index (Tables 4 and 5).

Most BAM is mainly based on high-resolution imagery (e.g., Landsat-8, Sentinel-2) for the normalized burned ratio index. Although this index has provided some promising results for BAM, due to the dependency of burned areas on the environmental features and the behavior of the fire, it is hard to discriminate burned areas from the background. This issue has reduced the efficiency of the BAM methods by the need to threshold the normalized burned ratio indices. Furthermore, some high-resolution burned area products on a global scale have been obtained based on this. Thus, these products do not support accurate BAM in practical real-world burned area estimation. Similarly, some unsupervised thresholding methods have been used, but due to the complexity of the background and noise conditions, the selection of suitable thresholds is another limitation of these methods.

Many BAM methods have been proposed based on machine-learning methods, such as random forest, K-nearest neighbor, and support machine vector. While these methods have provided acceptable results for BAM, they use handcrafted feature extraction. Manual feature extraction and then selection of suitable features is a time-consuming process. This issue needs to be considered when the study area is very large scale and the number of features is high. Additionally, these methods mainly focus on spectral features and ignore the potential of spatial features. The potential of spatial features has been shown in many studies on BAM based on machine-learning methods. The deep-learning-based methods can automatically extract deep features that are a combination of spectral and spatial features. This study has used the deep-feature extraction manner for BAM based on convolution and morphological layers.

To show further the effectiveness of the DSMNN-Net process, we visualized the feature maps in the different layers to look inside their internal operation and behavior. Figure 11 illustrates the visualization of the feature maps extracted from some layers in

the proposed DSMNN-Net for random pixels. The first layers show the shallow features, while the middle layers focus on the general structure around the central pixel.

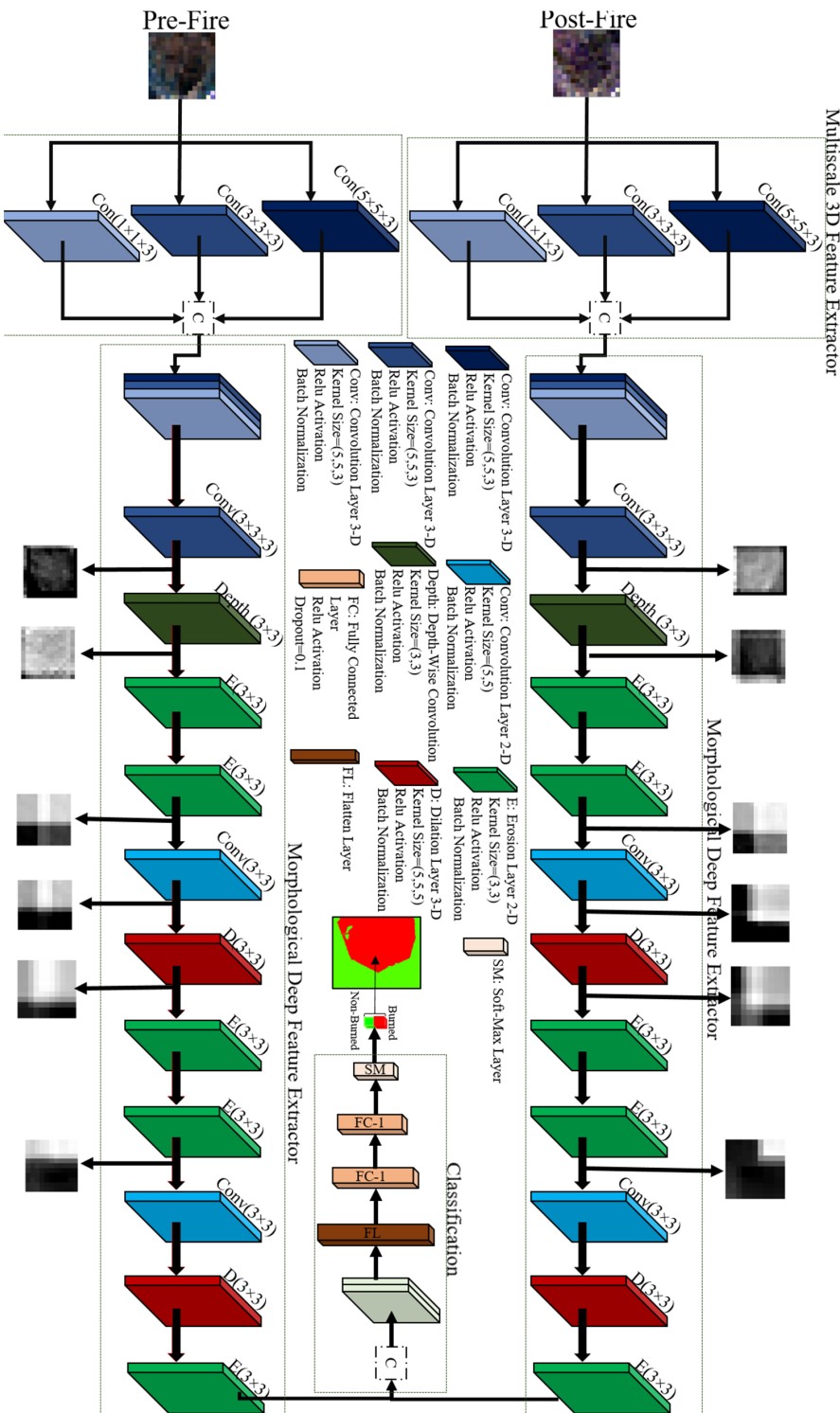

**Figure 11.** Visualization of feature maps in the DSMNN-Net for a random pixel.

The efficiency of deep features and handcrafted features can be seen in Table 6. Based on these data, the DSMNN-Net provided greater accuracy compared to other state-of-the-art methods for BAM using the Sentinel-2 imagery.

**Table 6.** Comparison of performance of the DSMNN-Net with other burned area mapping methods.

| Reference | Accuracy | Method | Dataset |
|---|---|---|---|
| Grivei, et al. [77] | (F1-Score: 0.873) | Support vector machine algorithm and spectral indices, factor analysis | Sentinel-2 |
| Barboza Castillo, et al. [78] | 94.4 | Thresholding on the spectral index | Sentinel-2 |
| Syifa, et al. [79] | 92 | Support vector machine and imperialist competitive algorithm | Sentinel-2 |
| Quintano, et al. [80] | 84 | Spectral index and thresholding | Combination of Landsat-8 and Sentinel-2 |
| Ngadze, et al. [81] | 92 | Random forest | Sentinel-2 |
| Roy, et al. [82] | 92 | Random forest change regression, and a region growing manner | Combination of Landsat-8 and Sentinel-2 |
| Lima, et al. [83] | 96 | Thresholding on the spectral index | Sentinel-2 |
| Seydi, Akhoondzadeh, Amani and Mahdavi [10] | 91 | Spectral and spatial features and random forest | Sentinel-2 |
| DSMNN-Net | 98 | Deep-learning based | Sentinel-2 |

OA, overall accuracy.

Hyperspectral imagery has a high content of spectral information in comparison with multispectral imagery. This advantage of hyperspectral imagery helps to detect burned areas with a highly complex background. Thus, the main reason behind the robust results provided by the method proposed here is the use of the hyperspectral dataset for the BAM. The burned pixels have a high similarity to some unburned pixels, and to clarify this subject, we presented some spectral signatures of burned and unburned pixels in the different areas. Figure 12 illustrates the similarities of the spectral signatures for the two main classes. Based on these data, the burned and unburned pixels have similar behaviors in the 0.45 to 0.8 μm range, while for other areas there are some differences in the reflectance. Therefore, hyperspectral imagery and combining pre-event datasets can be useful for BAM.

Additionally, the proposed suitable deep-feature extraction framework is very important in deep-learning-based methods. Among these three deep-learning-based methods, the method proposed here provided the best performance in all of the scenarios and for both study areas. This issue originated from the architecture of the deep-learning methods in the extraction of deep features. The DSMNN-Net extracts the deep features based on the type of kernel convolution and morphological layers. Initially, the DSMNN-Net uses multiscale 3D kernel convolution that investigates the relation among the spectral bands in deep-feature extraction. Based on Figure 12, there are some differences among the spectral signatures for the same classes, and these differences are greater for some spectral bands. Furthermore, there is some overlap between the two classes in the spectral bands. Therefore, using 3D convolution can enhance the efficiency of the network, because this can consider the relations between the spectral bands and the relations between the central pixel and the neighboring pixels. This advantage is the most important factor in taking the full content of spectral information for the BAM. Then, the morphological layers are used to explore nonlinear characterizers of the input dataset in the mapping. Thus, the DSMNN-Net can extract high-level and informative deep features based on proposed architectures; as a result, accurate BAM is possible using this proposed method. Additionally, the diversity of objects and the complexity of unburned areas mean that the BAM changes, which is a challenge. Solving this challenge mainly requires increasing the depth of the network, which results in an increasing number of parameters, and the need for greater training data and time. Here, the DSMNN-Net uses morphological operation cases to investigate the

complexity of the background. The morphological operators use nonlinear operations that increase the efficiency of the network for the BAM.

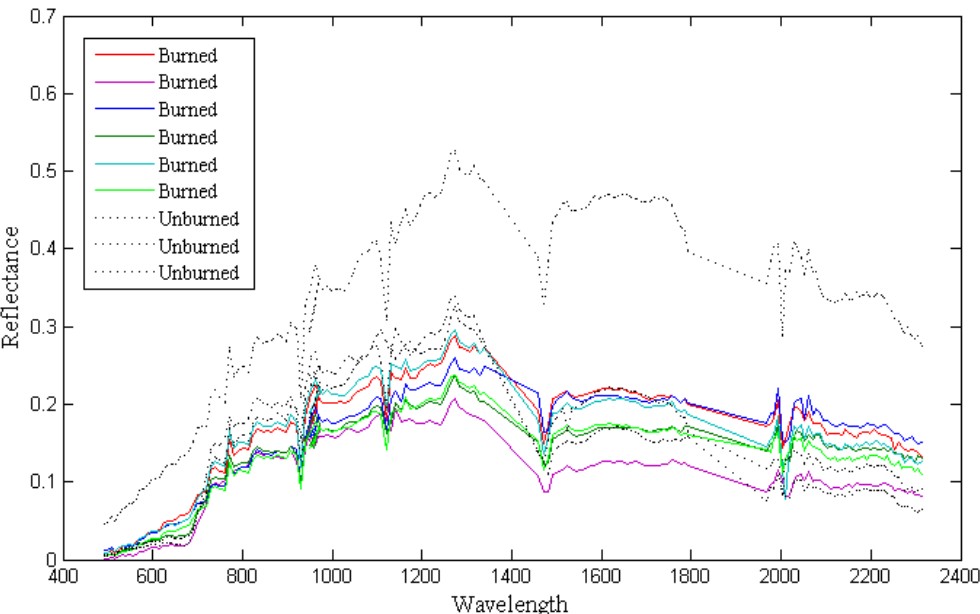

**Figure 12.** Spectral signatures among the burned and unburned pixels.

Recently, some semantic-segmentation-based (U-Net architecture) methods have been proposed. While these methods can provide considerably improved results for BAM, it needs to be noted that they require large amounts of labeled data of a specific size (i.e., 512 × 512 or 256 × 256). Obtaining a large amount of labeled datasets for such an application is very difficult and time consuming. Furthermore, these models are more complex due to the higher number of parameters and the need for more time for training the network. The DSMNN-Net used close to 45,000 pixels for the BAM, and obtaining this amount of sample data is easy according to the extent of the study areas.

One of the most common challenges for BAM based on changes in detection methods is the detection of nontarget changes. For example, the second study area has some nontarget change areas where their changes originated from changes in the water level of the lakes. This issue meant that the methods considered these as burned areas, while they are nonburned areas. This challenge is more evident in the BAM by the CNN method proposed in [42]. The sample data should cover more areas in the background, although the method proposed here controlled this issue in the BAM.

The method proposed here uses adaptive heterogeneous datasets in the mapping of burned areas. However, pixel-based change detection methods can be applied for bi-temporal multispectral/hyperspectral datasets easily, while they are difficult to apply for a heterogeneous dataset. In other words, some methods (i.e., image differencing algorithm) compare the pixel-to-pixel of the first and second time of bi-temporal dataset for BAM, while for heterogeneous datasets this is very difficult due to difference in a number of spectral bands, and content of datasets. The proposed DSMNN-Net can be applied to heterogeneous datasets without any additional processing (e.g., dimensional reduction). These advantages will also help in BAM when applied in a near real-time manner. It is worth noting that the proposed DSMNN-Net applied based on pre-event multispectral and post-event hyperspectral datasets while the bi-temporal pre-event and post-event hyperspectral datasets can improve the result of the BAM.

## 6. Conclusions

Accurate and timely BAM is the most important factor in wildfire damage assessment and management. In this study, a novel framework based on a deep-learning method (DSMNN-Net) and the use of bi-temporal multispectral and hyperspectral datasets was proposed. We evaluated the performance of the DSMNN-Net for two study areas in two scenarios: (1) BAM based on bi-temporal Sentinel-2 datasets and (2) BAM based on pre-event Sentinel-2 and post-event PRISMA datasets. Furthermore, the results for the BAM are compared with other state-of-the-art methods, both visually and numerically. The results of the BAM show that the method proposed here has high efficiency in comparison with the other methods for BAM. Additionally, the use of hyperspectral datasets can improve the performance of BAM based on deep-learning-based methods. The experimental results of this study illustrate that the DSMNN-Net has some advantages: (1) it provides high accuracy for BAM; (2) it has a high sensitivity for BAM for complex background areas; (3) it is adaptive, with heterogeneous datasets for BAM (multispectral and hyperspectral); and (4) it can be applied in an end-to-end framework without any additional processing.

**Author Contributions:** Conceptualization, S.T.S. and M.H.; methodology, S.T.S.; writing—original draft preparation, S.T.S.; writing—review and editing, S.T.S., M.H. and J.C.; visualization, S.T.S.; supervision, M.H. and J.C.; funding acquisition, J.C. All authors have read and agreed to the published version of the manuscript.

**Funding:** This study received no external funding.

**Institutional Review Board Statement:** Not applicable.

**Informed Consent Statement:** Not applicable.

**Data Availability Statement:** Publicly available datasets were analyzed in this study. These datasets can be found at https://scihub.copernicus.eu/, accessed on 16 November 2021 and http://prisma.asi.it/missionselect/, accessed on 16 November 2021.

**Acknowledgments:** The authors would like to thank the European Space Agency and the Italian Space Agency for providing the datasets. We thank the anonymous reviewers for their valuable comments on our manuscript. This research was partially supported by the AXA fund for research and by MIAI @ Grenoble Alpes, (ANR-19-P3IA-0003).

**Conflicts of Interest:** The authors declare that they have no conflict of interest.

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
