# Peer review of "DSMNN-Net: A Deep Siamese Morphological Neural Network Model for Burned Area Mapping Using Multispectral Sentinel-2 and Hyperspectral PRISMA Images"

_remotesensing, doi:10.3390/rs13245138_

Round 1

Reviewer 1 Report

1.  The author should introduce the research progress by their way, not just lists literatures one by one. It is particularly obvious in the introduction that this part needs to be reorganized.
2. There is a problem with the language, which needs to be carefully revised. The formula and description of the article and the article format also need to reorganize again.
3 the selection of sample data in Figure 6 is weird. Are there any selection rules for samples?
4. The method provided in this manuscript can be used to extract the  burning area, but if other conditions lead to vegetation condition transformation (such as felling), will this transformation be classified as the burning area?
5. The algorithm proposed by the authors makes use of Sentinel 2, What are the advantages of this data? Sentinel 2's return period is 10 days, can't it form real-time observation? Moreover, there are problems in business.
6. The authors lacks the analysis of the image spectral reflection on fire burned area. Therefore, when there are other interferences, the analysis results will be affected.

Author Response

Please see the attached response file.

Reviewer 2 Report

Tracking wildfire using hyperspectral and high-spatial-resolution satellite data is promising. This paper combined deep learning methods and these data to estimate burned area, which is interesting. However, some issues must be solved and clarified. My concerns are as follows:

  1. Abstract: what is a big challenge for existing methods to map burned area?
  2. Abstract: how did the authors’ method combine sentinel and hyper-spectral data?
  3. Abstract: show some prospects of the study in the end of abstract?
  4. Why did not the authors use the pre-event PRISMA data?
  5. Sentinel-2 and PRISMA have different spatial resolution. How did the authors process this issue?
  6. How did the authors acquire the ground truth data?
  7. Topography can affect surface reflectance (Proy et al., 1989; Hao et al., 2018) and thus the burned area mapping. Some discussion about the topographic effects can be added.

Proy, C., D. Tanre, and P. Y. Deschamps. "Evaluation of topographic effects in remotely sensed data." Remote Sensing of Environment 30.1 (1989): 21-32.

Hao, Dalei, et al. "Modeling anisotropic reflectance over composite sloping terrain." IEEE Transactions on Geoscience and Remote Sensing 56.7 (2018): 3903-3923.

  1. How did the authors make sure the dependence of the validation data compared to the training data?
  2. For the same spectral band, Sentinel-2 and RPISMA may have different surface reflectance values. How did the authors process this issue?

Author Response

Please see the attached response file.

Reviewer 3 Report

The manuscript proposed a DSMNN-Net, which integrated multi-scale convolution layers and erosion/dilation layers in CNN-Nets to detect the burn area by RS images. The authors illustrated that DSMNN-Net can be trained using fewer samples (45,000 pixels Line 505-507) in comparison with commonly used CNN models. The statement of the article is barely smooth, and the logic of the writing is less rigorous. I had some concerns for authors to consider.

1, Accordingly, It seems the novelty of this case study is the addition of morphological layers, but the authors did not give any interpretable reasons of why to so can improve the burned area mapping. I would expect to see some Intermediate visual results of solid explanations.

2, In Figure 2, the model ended with two fully connected layers, then followed by a soft mask layer, such an ending structure is generally utilized for image recognition (such as the computer vision task: there is a cat in the pic, so the pic was classified as the cat ) instead of image segmentation, am I right? This model is incomprehensible, how does a structure without upsampling layers (deconvolution layers) can achieve pixel-wise segmentation results?  Could you explain?

3, I did not see detailed descriptions of training data (it was mentioned once in the paper, line 276). Generally, hundreds of labeled images are needed to train a deep CNN model. Did you cut the ground truth to chips? How many chips? If not, please give some hints.

4, line 505-507: The DSMNN-Net has utilized under 45,000 pixels in the BAM and obtaining this amount of sample data is easy according to extent of the study area. How and where did you use these 45,000 pixels in the paper? Use them for training or validation? Clarifying.

5, Where is the validation dataset? Did you describe them? Or do I miss them? Give me some hints.

6, The conclusion is too optimistic, the article does not have more experiments to prove that the proposed model is effective for heterogeneous datasets, this is a just case study in AUS.

A large number of grammatical errors had affected the readability of the paper. Please utilize widely used Arabic numerals (1,2,3,4,5,6,7…) as the reference in line numbers.

Line 54-125, This paragraph needs to be more summarized instead of listing each case study individually. Organize them into several aspects in terms of works of literature.

Line 274, initial values? Do you mean random initialization? Or other pre-trained values. Clarifying it.

Line 290, Delete this sentence. It’s meaningless.

Line 294-295, Jaccard index is commonly named as Intersection over Union in computer vision. It’s better to also mention IOU.

Figure 7, It seems (c)’s result is worse than (a) and (b), in which a large number of unburn patches (green) were surrounded by burned patches (red).

Figure 9, (c)’s result is also worse than (a) and (b),

Author Response

Please see the attached response file.

Reviewer 4 Report

In this paper, authors proposed a new framework (DSMNN-Net) to deal with burned area mapping by using RS datasets. They showed improvement on the accuracy with comparison of previous literature. I think the paper is written well and is suitable, but there are some details missing which needs to be addressed. Here are my comments:

Some details in methodology section needs to be added:

  1. Preprocessing procedure is not clear in section 2. Any type of normalization or data cleaning should be stated
  2. How data splitted to train, validation and test datasets. What is the data distribution of each datasets?
  3. Section 2.1 ; line 3: is it “deep future extraction …“ or “deep feature extraction …” ? there are some other type of typos in the next subsection such as “coevolution layer” which needs to be corrected.
  4. More discussion on network architecture needs to be added. For example , Multi-scale 3D feature extractors layers needs to be explained in more details and the way the authors selected those three layers. Also, one or two line explanations about the importance if erosion layers needs to be added
  5. Batch size and other tuning parameters needs to be stated clearly and the way they were selected. For example, why number of epochs are 500?

Also, there are some comments on the results and discussion part:

  1. It was stated in section 3.3 that 169 spectral bands are selected for the analysis. Put the starting and final wavelengths with the procedure for removing noise.
  2. Did you select any outlier detection procedure?
  3. The definition of scores and accuracy are not clear? Why the authors didn’t use IoU or mAP for measuring the accuracy?
  4. A comparison of this procedure with previous method such as Unet is not complete. Unet can be applied to small amount of data as well and it is easy to use pre-trained data of Unet from other datasets. I think the authors need to compare the models based on the number of parameters, complexity and other features as well

Author Response

Please see the attached response file.

Round 2

Reviewer 1 Report

Line 346-348-353 there is a problem that a sentence becomes an independent paragraph. It's best to describe them in one paragraph.

Author Response

Please see the attached response file.

Reviewer 2 Report

The authors tried their best to solve my concerns. This verson looks good. I just have one suggestion:

If the pre-event hyperspectral data is available, the model can be improved further I believe. The authors can add this in the discussion part.

Author Response

Please see the attached response file.

Reviewer 3 Report

No further comments.

Author Response

Please see the attached response file.

Reviewer 4 Report

I like the revised version. Great job. 

Author Response

Please see the attached response file.
